# Use of the Ferroelectric Ceramic Bismuth Titanate as an Ultrasonic Transducer for High Temperatures and Nuclear Radiation

**DOI:** 10.3390/s21186094

**Published:** 2021-09-11

**Authors:** Brian T. Reinhardt, Bernhard R. Tittmann

**Affiliations:** Department of Engineering Science and Mechanics, Penn State University, University Park, PA 16802, USA; nairbt85@gmail.com

**Keywords:** high temperature, ultrasonic testing, radiation tolerance, nuclear reactors, gamma radiation, transducer, sensors

## Abstract

Ultrasonic transducers are often used in the nuclear industry as sensors to monitor the health and process status of systems or the components. Some of the after-effects of the Fukushima Daiichi earthquake could have been eased if sensors had been in place inside the four reactors and sensed the overheating causing meltdown and steam explosions. The key element of ultrasonic sensors is the piezoelectric wafer, which is usually derived from lead-zirconate-titanate (Pb(Zr, Ti)O_3_, PZT). This material loses its piezoelectrical properties at a temperature of about 200 °C. It also undergoes nuclear transmutation. Bismuth titanate (Bi_4_Ti_3_O_12_, BiTi) has been considered as a potential candidate for replacing PZT at the middle of this temperature range, with many possible applications, since it has a Curie–Weiss temperature of about 650 °C. The aim of this article is to describe experimental details for operation in gamma and nuclear radiation concomitant with elevated temperatures and details of the performance of a BiTi sensor during and after irradiation testing. In these experiments, bismuth titanate has been demonstrated to operate up to a fast neutron fluence of 5 × 10^20^ n/cm^2^ and gamma radiation of 7.23 × 10^21^ (gamma/cm^2^). The results offer a perspective on the state-of the-art for a possible sensor for harsh environments of high temperature, Gamma radiation, and nuclear fluence.

## 1. Introduction

Current research and development are targeting the radiation endurance of transducers for possible sensors in reactors [1]. Bismuth titanate (BiTi) has been viewed favorably because of its potential to replace ferroelectrics such as lead-zirconate-titanate (Pb(Zr, Ti)O_3_, PZT) for application in high-temperature and radiation environments. This paper describes the details and conclusions of a long-term test of bismuth titanate (BiTi) in the high-temperature and nuclear radiation environment of a nuclear test reactor.

## 2. Bismuth Titanate (BiTi)—Material Properties

Bismuth titanate (BiTi) was discovered in 1949 by Bengt Aurivillius [2]. It is a bismuth layered ferroelectric oxide that belongs to the Aurivillius family—structures that are described by the formula (Bi_2_O_2_)2+ (M*_n_* − 1R*_n_*O_3_*_n_*
_+_ _1_)2−, where *n* is 1 to 6. The (M*_n_* − 1R*_n_*O_3_*_n_*
_+_ 1)2− formula is the *n* pseudo perovskite unit which is in between two (Bi_2_O_2_)2+ layers, as shown in Figure 1. In the case of bismuth titanate (BiTi), where *n* = 3, the M cation (typically a large mono-, di-, or tri-valent cation) is Bi_3+_, and the R cation (a smaller tri-, tetra-, penta-, or hexa-valent cation) is Ti_4+_.

Bismuth titanate (BiTi) was originally prepared by ball milling a combination of Bi_2_O and TiO_2_ powders, then sintering at a very high temperature. This led to an accumulation of large Bi_4_Ti_3_O_12_ particles, causing poor microstructure and ferroelectric properties. To prevent this from happening, chemical precursor methods are now used, including co-precipitation, hydrothermal, and molten salt synthesis [3,4,5,6,7].

Bismuth titanate has a plate-like microstructure with anisotropic properties, a low coercive field (E*_c_*), small remnant polarization (P*_s_*), high dielectric strength, high dielectric constant (approximately 200), and good fatigue properties. Its high Curie temperature [3] allows it to be used for high-temperature piezoelectric applications (>300 °C), memory storage, and optical displays [6,7,8,9]. BiTi-100, produced by Del Piezo Specialties, LLC, has a high Curie temperature and high piezoelectric properties, as shown in Table 1 listing the most relevant material properties [8].

Kazys et al. and Ferrandis et al. [10,11,12,13] have discussed the effects of nuclear radiation on bismuth titanate and shown that it is resistant to low-level gamma and neutron radiation.

## 3. Overview of Reactor Project

In 2012, a research grant was awarded to the Penn State University (PSU) by the Advanced Test Reactor—National Science User Facility (ATR-NSUF) to work with Idaho National Laboratory, Pacific Northwest National Laboratory, Argonne National Laboratory, and the Massachusetts Institute of Technology to develop and test piezoelectric and magnetostrictive transducers for in-pile applications. This paper is based on the Penn State Ph.D. Thesis of Dr. Brian Reinhardt [14] and will discuss the details of the experiment for bismuth titanate (BiTi). The results for the other sensor materials have been discussed in detail elsewhere [15,16,17,18,19,20,21].

A capsule was designed to house the ultrasonic sensors for an in-pile test at the Massachusetts Institute of Technology Reactor (MITR) and was inserted into the reactor on 18 February 2014 and removed on 12 May 2015. The samples were in the reactor for a total of 448 days before they were moved to the hot cell for post-irradiation analysis. The capsule experienced scheduled and unscheduled shutdowns as it was riding along with the reactor’s typical operation. Due to periodic shutdowns, the capsule was only exposed to 219 days with the reactor above 5 MW.

## 4. Reactor Environment

The MITR is a heavy-water-reflected, light-water-cooled, and moderated nuclear reactor that uses plate-type fuel packed into a rhombohedral shape. An artistic rendering of the MITR is shown in Figure 2 and the rhombohedral fuel elements are noticeable.

Figure 2 is a cartoon of the reactor pressure vessel. The blue tube is a dry tube with the ability for atmosphere regulation and is where the capsule for this experiment was located. MIT used helium and neon to control the temperature during the experiment.

The purpose of the experiment was to expose the transducers to a fast fluence, with neutrons with a higher energy than 1 MeV, of approximately 10^21^ n/cm^2^. To achieve this, the top of the reactor was lifted off and the capsule was lowered through the dry tube into position, near the peak flux produced by the reactor fuel elements. The capsule housed each of the sensors and provided cabling so that they could be monitored in a safe environment. The cables inside the reactor were selected for their radiation tolerance. They were connected to a breakout box designed to impedance-match the cables that would run outside the reactor to the monitoring equipment. The experimental capsule was placed in the blue tube, indicated by the red arrow in Figure 3.

## 5. Capsule Environmental Characterization

Due to space limitations, only a finite number of sensors could be tested. Here, we report only on bismuth titanate (BiTi). The other instruments placed in the capsule were used to monitor environment variables. Figure 4 shows a 3D rendering of the ultra-capsule as constructed for the irradiation. The bismuth titanate (BiTi) sensor is shown in yellow [14].

A cable fixture was designed to hold all cables in position and alleviate strains as the capsule was lowered into position in the reactor.

In order to monitor temperature, two K-type thermocouples were used. These have been used extensively in reactor environments and have a minimal drift, 2%, at temperatures above 850 °C and exposed to neutron fluence of up to 2 × 10^22^ n/cm^2^ [22]. The temperature measured by these two thermocouples is shown in Figure 5.

In addition to the thermocouples, the maximum temperature reached at selected locations during the irradiation was verified using melt wires encapsulated in a quartz tube. Five wire compositions were included, with melting temperatures ranging between approximately 327 and 514 °C. Melting temperatures of the wires have been verified using a Differential Scanning Calorimeter (DSC). Integrated thermal and fast neutron fluences at selected locations have been evaluated through post-irradiation analysis of Fe-Ni-Cr flux wires.

Two sensors were placed in the capsule to monitor both gamma and neutron flux. The gamma flux detector was a platinum emitter, self-powered gamma detector, while the neutron detector was a vanadium emitter, self-powered neutron detector. The basic principle behind both detectors was that the radiation either transmutes species or induces radiolysis to generate small electrical currents depending on the flux levels.

## 6. Neutron Power

Neutron power is proportional to the overall fission rate, while thermal power measures the heat generated by the reactor. While the fission was driving the overall thermal process, alternative sources of heat were generated from gamma radiation. During the experiment, the reactor was monitored and controlled by the position of six boron-stainless-steel blade-type control rods.

Depending on the needs, the reactor may be required to be shut down without much notice. These events are called SCRAM (Safety Control Rod Axe Man) events and could happen at any time, reducing the neutron flux and temperature of the experiment. There are several outages that were scheduled for various reasons, such as refueling, testing, or adjusting experimental equipment. The reactor power was measured by the MIT staff and provided along with the thermocouple output, flux detector output, and a time stamp for reference with the acoustic data. The reactor neutron power for this experiment is plotted in Figure 6.

During the experiments, it became convenient to discuss the samples’ performance during specific power cycles. As such, a power cycle was defined as any stretch of time that the reactor is at constant power, not including SCRAM events. These power cycles are indicated in Figure 6.

### 6.1. Neutron Fluence

For this experiment, the neutron fluence was determined from the theoretical neutron flux at a reactor power of 5.8 MW. A linear assumption was made with the flux at full power and the flux at intermediate power levels.

### 6.2. Gamma Fluence

The total gamma flux was 3.2 × 10^23^ gamma/cm^2^-s at 5.5 MW. The flux at intermediary power levels was approximated using a linear relationship between gamma flux and neutron power. The max gamma exposure was approximately 7.23 × 10^21^ gamma/cm^2^.

## 7. Measurement Process

### 7.1. Insertion Capsule

An insertion capsule with a piezoelectric transducer was designed for the tests and fabricated with bismuth titanate (BiTi) as the active element (BT-100) [8]. Figure 7 and Figure 8 show a sketch and a photo, respectively, of the insertion capsule. High-purity aluminum foil was used to bond the bismuth titanate sensor to a Kovar cylinder used as an ultrasonic waveguide for a pulse-echo operation.

Spring pressure was used for coupling the bismuth titanate wafer to the waveguide. The backing material was a layer of carbon-fiber/carbon-matrix composite (C/C, obtained from Bendix Airbrake Co.). After the sensor was assembled, the test capsule was heat-treated at 610 °C to soften the foil and achieve better coupling to the waveguide. The capsule was then readied for insertion into the reactor. Included in the reactor insertion was a “drop-in” bismuth titanate wafer for post-treatment material examination.

First, the test capsule with the bismuth titanate wafer was tested up to 650 °C and was found to be operating well up to 550 °C. The pulse-echo amplitude experienced a significant increase in amplitude at 200 °C, rising as much as 250%. This was thought to be a result of softening of the coupling film providing better ultrasonic transmission. This amplitude was maintained until 550 °C. Upon cooling of the sample chamber, the ultrasonic pulse-echo amplitude returned to its pre-failure amplitude. The bismuth titanate always recovered its high pulse amplitude upon re-heating. The signals were analyzed in the frequency domain and were found to have several peaks in the frequency spectrum of the signal. Besides the main through thickness resonant mode at 2.1 MHz, up to 6 odd harmonics were found.

### 7.2. In-Pile Ultrasonic Measurement Equipment

During the irradiation, the sensors were hooked up to a national instruments NIPXI 1042 chassis through PXI-2593 switch cards. The switch card was connected to the NDT-5800 PR pulser/receiver and a SMX-2064 Digital Multimeter. This enabled either DC impedance measurements or pulser/receiver measurements. The settings used for the pulser/receiver are saved alongside each of the waveforms in an associated file. In this way, measurement parameters were accessible after the experiment. The pulser/receiver was connected to a ZTEC ZT4211 digital oscilloscope. Finally, everything was connected in LabVIEW 2009. The LabVIEW program was designed so that periodic measurements could be made on each of the transducers connected to the switch board. Custom software to control the equipment was developed by Data Science Automation, Inc., utilized during the irradiation. The data output was organized by the transducer and tagged with the acquisition date and time.

### 7.3. Out-of-Pile Optical Inspection and Ultrasonic Measurements

After the irradiation, the capsule was moved into one of MIT’s hot boxes. This box provides sufficient shielding to view and operate on materials from the reactor. Due to the dangerous levels of gamma emission from the capsule, it was not possible to handle any of the components without proper caution. In the hot box, manipulators were used to interact with the capsule and deconstruct various components. An off-the-shelf web camera was dropped into the hot box and used to both record the deconstruction process as well as take photographs at various angles. Pictures of the various components were taken using this webcam.

Pulse-echo measurements were performed after the irradiation. The first time was while the sensors were still connected to the LabVIEW box and thus the measurement procedure did not change from those described in the in-pile measurement procedures. After the sensor was deconstructed, and the piezoelectric inserts were removed from the capsule, their activity was measured. If it was low enough for transportation, they were moved to a measurement area shielded with lead bricks.

To conduct acoustical measurements on these samples, a fixture was designed and fabricated from PVC, as shown in Figure 9. A female–male 50 Ω BNC adapter was screwed into the PVC. One of the nickel plungers, used during the construction of the piezoelectric sensors, was inserted into the female end of the BNC adapter and a BNC cable was attached to the other end of the adapter.

Since the sensor material was not attached to a waveguide, a waveguide was constructed to be one inch in diameter and one inch in length, cut from aluminum 6061. The waveguide was placed underneath the nickel plunger in the PVC fixture. Water was used to couple the sensor material to the waveguide. A small drop was placed on the surface of the waveguide, and the sensor was placed on the drop. A small piece of aluminum foil was placed as a lead conductor on the sensor material and the nickel plunger was lowed to contact the lead electrode and apply pressure to the sensor material.

The other end of the BNC cable was attached to the LabVIEW measurement system, which was used to pulse/receive to and from the sensor.

## 8. Out-of-Pile Measurements of Piezoelectric Coefficient, d_33_

The out-of-pile d_33_ measurements were performed with an APC International wide-range d_33_ which is shown in Figure 10. The d_33_ m can measure the range of 1–2000 pC/N with an accuracy of 0.1 pC/N. It used a piezoelectric actuator to mechanically actuate the specimen at 100 Hz with a force of 0.25 N.

Before each measurement, the system was calibrated according to the manual calibration procedures and provided reference specimen. Since the d_33_ is dependent on the force applied by the clamp system, a mark was placed on the dial to count the number of turns when applying pressure to the element. Since the specimens used were still too active, these measurements were conducted by the MIT staff utilizing lead brick shielding.

## 9. Data Processing

Although many measurements can be obtained from the pulse-echo response, the purpose of the experiment was to characterize the sensor materials’ performance throughout the experiment. As such, parameters such as pulse-echo amplitude, center frequency, and quality were measured where possible. The pulse-echo amplitude was measured as the peak of the fundamental harmonic. The center frequency was measured as the average of the −3 dB left and right frequency values. This was measured by first identifying the central peak, f_pk_, and subsequently measuring the frequency, where the power efficiency dropped by 50%. Approximately 5000 data points were obtained during the experiment, averaging out to be approximately ten measurements per day. This is approximately 30 GB of data, all of which needs careful organization. Before the data could be analyzed, they needed to be converted from a directory of excel files to a Matlab data file. Although the acoustic measurements were performed using the aforementioned National Instruments box and associated equipment, other equipment was utilized to monitor the thermocouples, self-powered detectors, and reactor power. These measurements were shared with us by MIT in several comma separated files with associated acquisition date and time stamps. Since all data were given date and time stamps, for each A-scan, the nearest environment time stamp to the acoustic measurement time stamp was used as the environment variable parameters for that measurement. This simply required a parsing program to extract the date from the file name and compare it with dates associated in the environment variable spreadsheet. Several programs were developed to handle the synthesis of the data, and most programs were developed using MatLab. To view the data on a day-to-day basis, a plotting tool was developed to access environment variables, monitor A-scan signals from each of the transducers, and select different processing techniques based on an adjustable window.

## 10. Bismuth Titanate (BiTi) Sensor Results

### 10.1. Ultrasonic Measurements

The bismuth titanate transducer was pulsed periodically using the LabVIEW system described in the in-pile ultrasonic measurement section. The pulser/receiver sampled the sensor output at 1 GS/s for a period of 100 µs. The pulser was set to a repetition rate of 1 kHz and a total of 256 measurements were averaged for each measurement. A total of 5038 measurements were obtained during the irradiation, with measurements being performed every 30 min until the first power cycle, when the sensor was sampled every 2 h.

The first waveform was acquired directly after the transducer was inserted into the reactor, as shown in Figure 11. There was a significant ring down time from the main bang, taking between 10 and 15 µs to subside. Figure 12 shows the Fast Fourier Transform (FFT) of A-scan collected directly after insertion of bismuth titanate (BiTi) into the reactor [14].

### 10.2. Performance during Irradiation

The performance of the bismuth titanate (BiTi) sensor was measured as the amplitude of the fundamental harmonic, measured using the power spectral density. The amplitude as a function of accumulated fluence is plotted in Figure 13.

An initial decline in the amplitude was observed starting on day 10 or at 1.28 × 10^19^ n/cm^2^, and reached a minimum relative amplitude of 60% at 5.29 × 10^19^ n/cm^2^ on day 25. The amplitude gradually recovered to a relative amplitude of 100% at 1.1 × 10^20^ n/cm^2^. None of the environment variables were associated with this change in amplitude. The relative amplitude of 100% was maintained until an accumulated fluence of 5 × 10^20^ n/cm^2^ was reached. Then, there was a significant decrease in the amplitude. A decline developed over a period of 17 days, dropping the pulse amplitude by 80%. Then, there was some recovery and some small amplitude remained until nearly the end of the experiment.

Figure 14 shows the A-scans measured at the beginning of the experiment, just before the decline, and just after the decline. There was a significant change in the time domain of the signal. An echo was observed just before 20 microseconds, which indicated that the sensor was still capable of transduction, however, its performance had significantly declined.

## 11. Bismuth Titanate (BiTi) Drop-In Sample

### 11.1. Overview

In case the bismuth titanate (BiTi) was not recoverable from the sensor capsule, an additional piece was placed in the test capsule to be irradiated. Although the resources at MIT were limited in terms of post-irradiation analysis, the activity of the bismuth titanate (BiTi) was low enough that it could be removed from the hot box. Several measurements with the sensor were reported: pulse-echo, impedance, and d_33_.

As shown in Figure 15, the bismuth titanate drop-in wafer was significantly darker after the irradiation. Aside from the discoloration, little visible differences were seen in the wafer.

### 11.2. Ultrasonic Measurements on Drop-In Sample of Bismuth Titanate

Figure 16 and Figure 17 show the case of pulse-echo from a piece of bismuth titanate in a pristine condition, as received from TRS Technologies, and the irradiated drop-in sample, respectively. Although the same ultrasonics pulser/receiver settings were used, there was a significant difference in the pulse-echo amplitude. However, the irradiated drop-in bismuth titanate active element did show a non-negligible signal after the full treatment in the reactor.

## 12. Measurements of Piezoelectric Coefficient, d_33_

A d_33_ m was purchased from American Piezo Ceramics, Inc., and used to measure d_33_ for each of the insert pieces. The meter was calibrated to a calibration block before performing measurements. The insert samples were safe to handle outside the hot cell, with no significant radiation only a few feet away. However, they were not safe to directly manipulate by hand and so, for safety, bricks were placed between the meter and the sensor block and the personnel. The d_33_ m reading and calibration depend on the force applied to the sensor. To control the force exerted by the meter of the piezoelectricity, an indicator mark was placed on the top of the dial. Once the pin made contact with the sensor, the dial was rotated 720 degrees. Using this procedure, calibration measurements were repeatable within ±1% of the reported value.

As can been seen from Table 2, post-irradiated d_33_ is approximately half of that of the pristine bismuth titanate. This is consistent with the reduction in pulse amplitude in the A-scan measurements of Figure 16 and Figure 17. On the other hand, these results show that bismuth titanate did survive the harsh reactor conditions, although with a considerable decline in piezoelectric performance once the threshold of flux 5 × 10^20^ n/cm^2^ was reached.

## 13. Radiation Damage

The results raise questions about the source of the damage incurred by bismuth titanate (BiTi) in these reactor experiments. It is apparent that considerable damage must have resulted from the effects of radiation. The effects of temperature and radiation on piezoelectricity have been studied to some extent, previously [23,24,25,26,27,28,29,30,31,32,33,34,35,36,37,38,39,40,41,42,43,44,45,46,47,48,49,50,51,52,53,54,55,56]. According to Parks et al. [56], since the temperature in the reactor was kept below the transition temperature (Curie temperature), the effects might be narrowed down to four primary forms of damage:De-poling via thermal spike processes;Amorphization/metamictization due to displacement spikes or high concentration of point defects;Increase in point defect concentration;Development of defect aggregates.

In the work of Parks et al. [56], two likely damage mechanisms were outlined, namely thermal spikes and displacement spikes. Additionally, transmutation products are considered, as these in fact induce both thermal spikes and displacement spikes in some cases. The radiation tolerance of piezoelectric materials is governed by not only resistance to de-poling in thermal spikes, but also resistance to amorphization, and a lack of atomic species having large nuclear cross-sections for thermal neutron-induced transmutations. Bismuth titanate (BiTi) contains bismuth (Bi), which has approximately double the cross-sections of titanium (Ti) or oxygen (O), approaching that of lead (Pb), as shown in Table 3. However, a thorough discussion of this topic is beyond the scope of this paper.

## 14. Conclusions

Developing ultrasonic transducers for in-pile applications is a growing concern in the nuclear industry. With the advances in liquid-cooled sodium fast reactors, development of high-efficiency nuclear fuels, and demands for increased security, having ultrasonic tools to investigate the evolution of the material microstructure and structural integrity is becoming a high priority. Considerable effort has been expended in the development of sensors based on single-crystal wafers, with good success, such as aluminum nitride [18,19,20,21]. However, many researchers have found value in working with ceramics, such as bismuth titanate (BiTi), because of the ease of preparation and the many means of preparation, such as sol-gel and spray-on processes. With the spray-on process, the ceramic can be deposited on curved surfaces, such as pipes, and used to generate guided waves [32,33,34,35,36,37,38]. It is therefore valuable to know about the survival of this ceramic in a harsh environment, such as in a nuclear reactor.

This paper summarized the testing of piezoelectric sensors with bismuth titanate (BiTi) active elements while being tested in a nuclear reactor for a period of 448 days. During this time, the sensors were subject to a neutron flux comparable to a commercial reactor. The sensors were subjected to high temperatures in the range of 400 to 450 °C. The sensor also saw constant fluctuations due to either SCRAM events or scheduled shutdowns, cycling the temperature environment.

The results showed that measurements of the sensors could be performed throughout most of the duration of the experiment. This is the first time that bismuth titanate (BiTi) has been tested to a fast fluence of 8.68 × 10^21^ (n/cm^2^) in an instrumented lead test. Table 4 shows the list of fluence metrics for the sensor, and Table 5 summarizes the performance of bismuth titanate (BiTi) during the experiment.

Although the performance of the sensors varied depending on the environment variables as well as the duration for which those variables remained constant, the last fluence level of a measurable echo from the back wall of the waveguide was reported. Further, the percent difference of this amplitude to the amplitude of the reflection at the beginning of the experiment, given similar environment variables, was also reported.

## Figures and Tables

**Figure 1 sensors-21-06094-f001:**
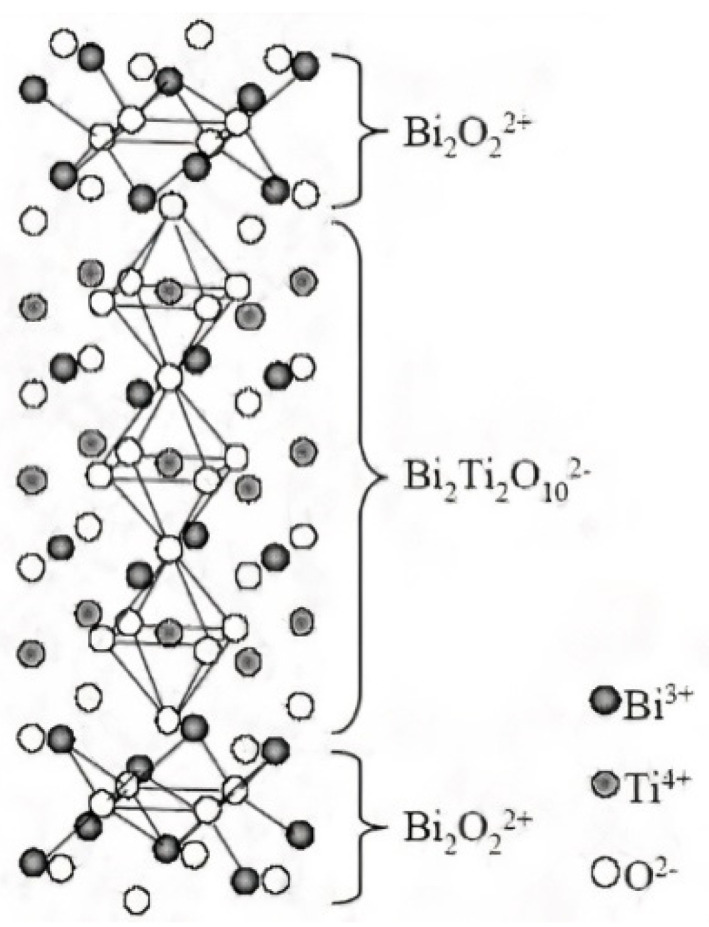
Crystal Structure of bismuth titanate (BiTi) [3].

**Figure 2 sensors-21-06094-f002:**
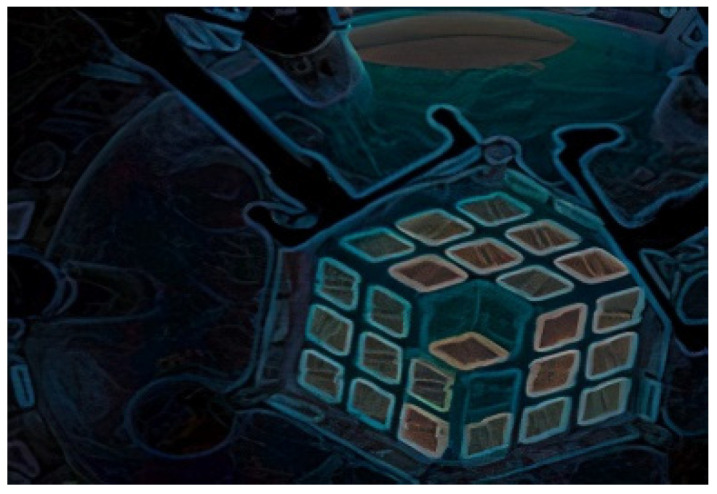
Artistic rendering of the MITR reactor core [14].

**Figure 3 sensors-21-06094-f003:**
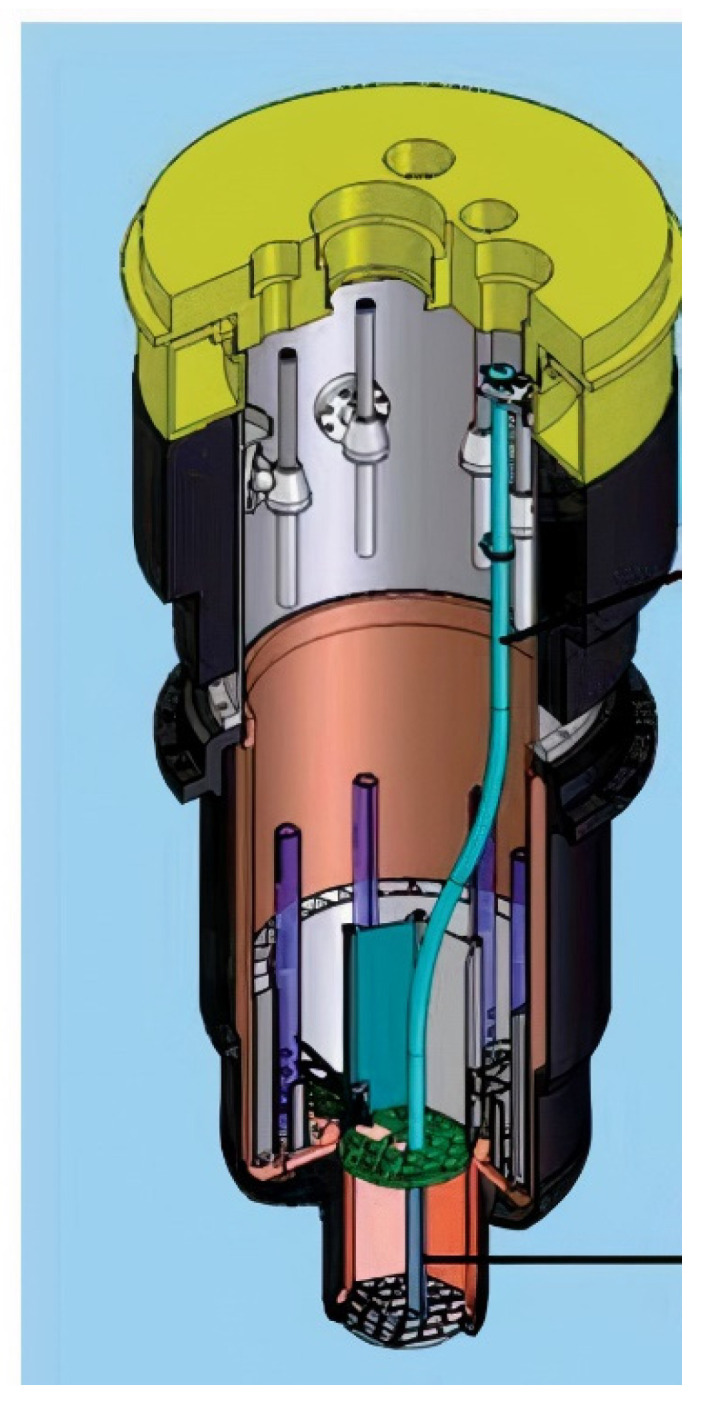
MITR pressure vessel [14].

**Figure 4 sensors-21-06094-f004:**
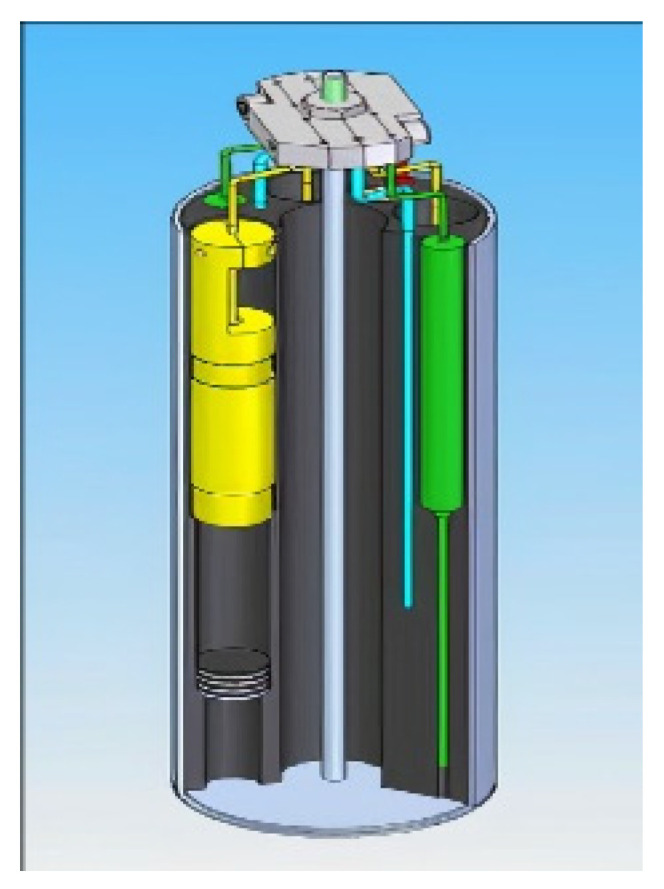
3D renderings of the ultra-capsule as constructed for the irradiation. The bismuth titanate (BiTi) sensor is shown in yellow [14].

**Figure 5 sensors-21-06094-f005:**
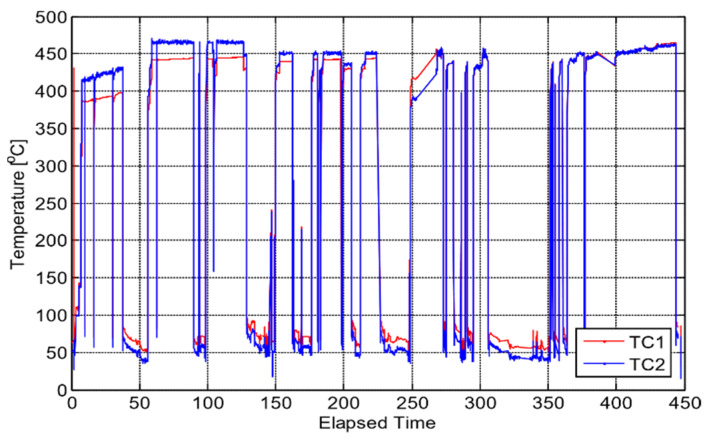
Temperature of bismuth titanate (BiTi) as a function of elapsed time (days) calculated from TC1 and TC2 thermocouples [14].

**Figure 6 sensors-21-06094-f006:**
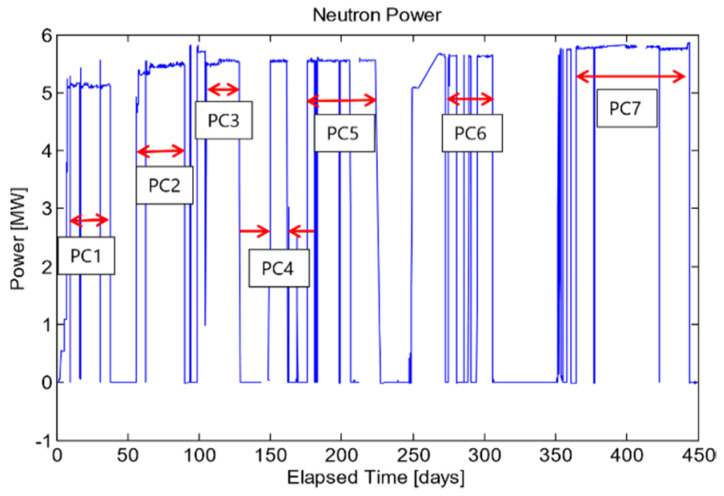
Plot of neutron power vs. elapsed time. PC# represents a full power cycle for which the neutron power remained relatively constant [14].

**Figure 7 sensors-21-06094-f007:**
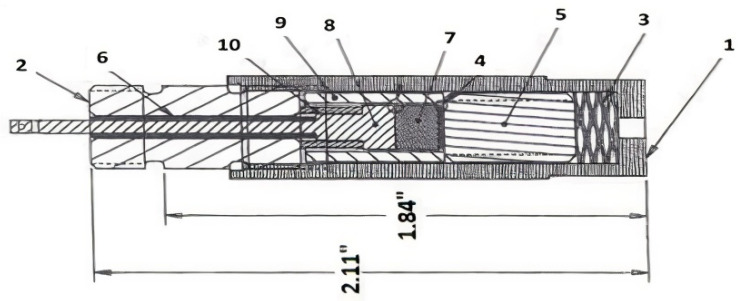
Sketch of the transducer capsule consisting of an outer casing (**1**), a cap (**2**), a high-temperature spring (**3**), the bismuth titanate wafer (**4**), the Kovar cylinder (**5**), the coaxial connection (**6**), the C/C backing material (**7**), the pressure plunger (**8**), and insulation (**6**,**9**,**10**). The bismuth titanate wafer was placed between the backing and the Kovar cylinder using foil [14].

**Figure 8 sensors-21-06094-f008:**
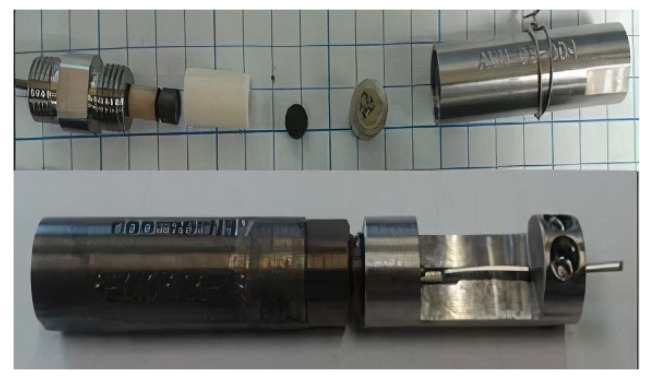
Photos of the test capsule components shown as a diagram in Figure 1 [14].

**Figure 9 sensors-21-06094-f009:**
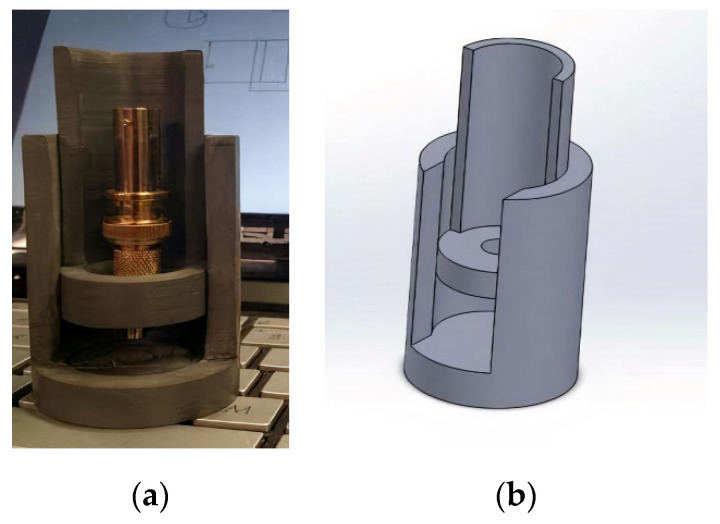
Sample fixture for pulse-echo measurements [14]. (**a**) On left is a photo and (**b**) on the right is the design sketch.

**Figure 10 sensors-21-06094-f010:**
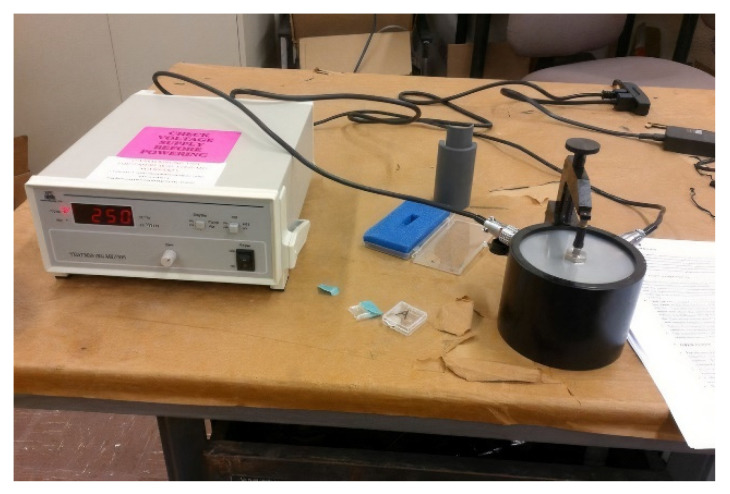
d_33_ Meter showing calibration block force reading [14].

**Figure 11 sensors-21-06094-f011:**
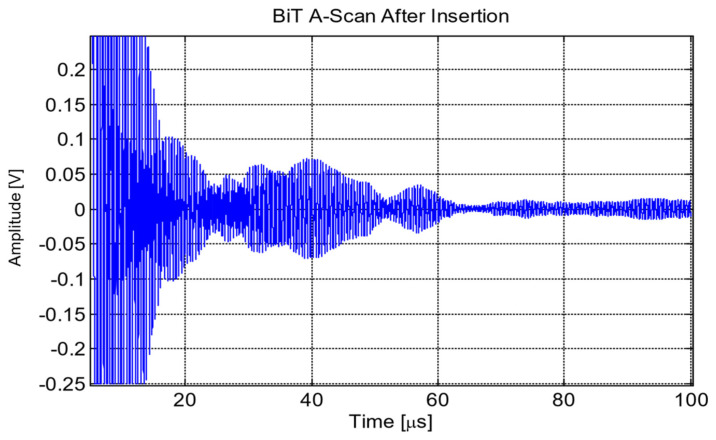
Bismuth titanate (BiTi) A-scan measured directly after insertion [14].

**Figure 12 sensors-21-06094-f012:**
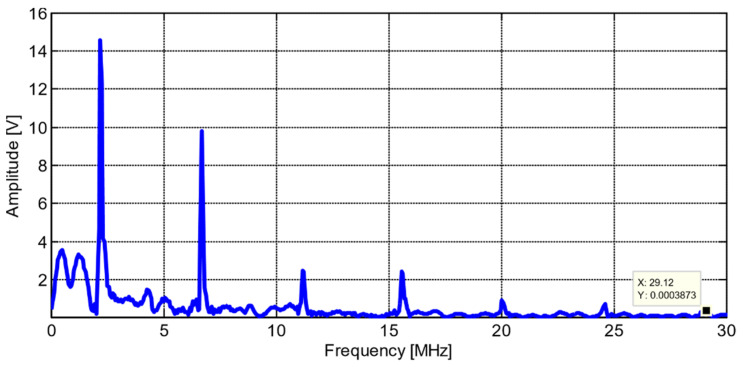
Fast Fourier Transform (FFT) of A-scan collected directly after insertion of bismuth titanate (BiTi) into the reactor [14].

**Figure 13 sensors-21-06094-f013:**
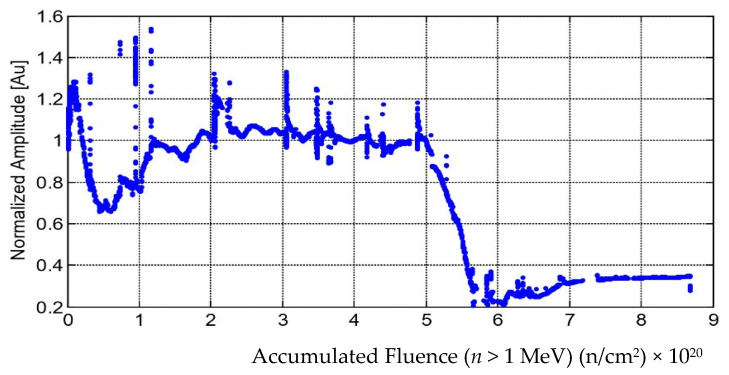
Bismuth titanate (BiTi) sensor pulse-echo amplitude of fundamental harmonic as a function of accumulated fluence. The spikes in amplitude occurred during periods where the reactor dropped to zero neutron power [14].

**Figure 14 sensors-21-06094-f014:**
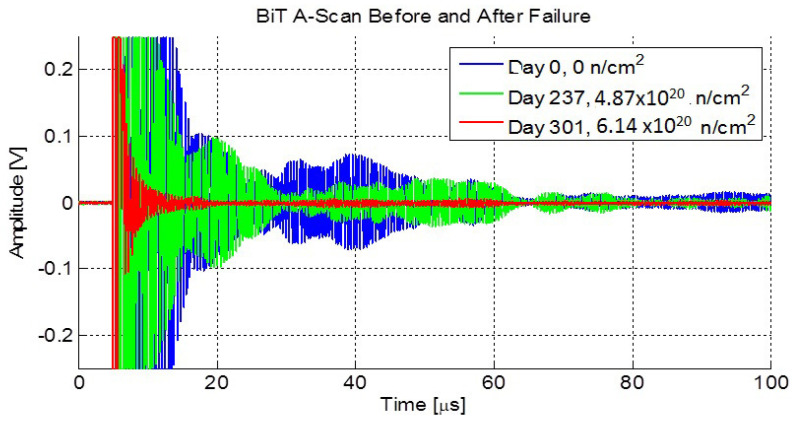
Comparison of A-scans (pulse-echo amplitude versus time) from days 1, 237, and 301 from the BiTi sensor. These represent A-scans from before and after the decline of the sensor [14].

**Figure 15 sensors-21-06094-f015:**
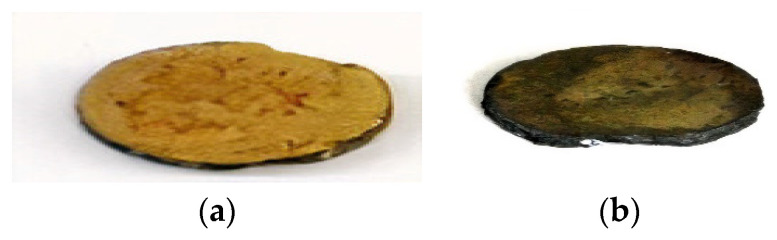
Bismuth titanate (BiTi) drop-in wafers. To the left in (**a**) is the pre-irradiated image and to the right in (**b**) is the post-irradiated bismuth titanate (BiTi) sample [14].

**Figure 16 sensors-21-06094-f016:**
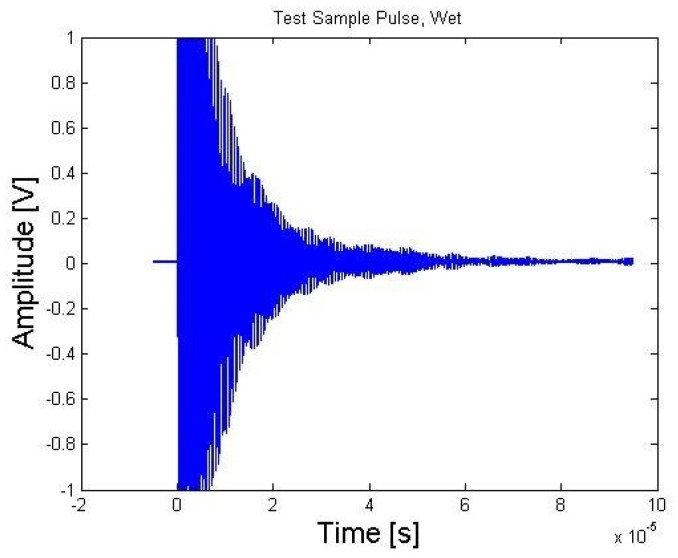
A-scan on an aluminum block using a pristine bismuth titanate active element [14].

**Figure 17 sensors-21-06094-f017:**
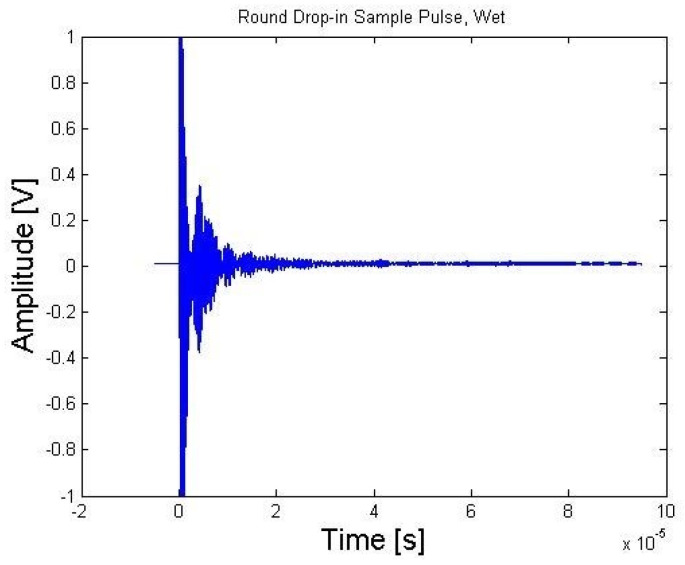
A-scan on an aluminum block using the irradiated drop-in bismuth titanate active element [14].

**Table 1 sensors-21-06094-t001:** Bismuth titanate (BiTi) material properties [8,14].

Material	Structure	Transition Temp (°C)	Transition Type	d_33_ (pC/N)	K_33_
Bi_4_O_12_T_3_	Perovskite	650	Curie Temp	20	0.23

**Table 2 sensors-21-06094-t002:** d_33_ readings of bismuth titanate sensors [14].

Post-Irradiated d_33_ (pC/N)	Pre-Irradiated d_33_ (pC/N)
10.8–11.3	20

**Table 3 sensors-21-06094-t003:** Cross-sections of materials.

Nuclei	Cross-Section Barnes (10^−24^ cm^2^)
Bismuth (Bi)	9.16
Titanium (Ti)	4.35
Oxygen (O)	4.23
Lead (Pb)	11.2

**Table 4 sensors-21-06094-t004:** Total fluence for irradiation in MITR [14].

Radiation Source	Exposure
Thermal flux (<0.4 eV)	2.27 × 10^20^ (n/cm^2^)
Epithermal flux (0.4 eV, 0.1 MeV)	1.72 × 10^21^ (n/cm^2^)
Fast flux 1 (>0.1 MeV)	1.88 × 10^21^ (n/cm^2^)
Fast flux 2 (>1 MeV)	8.68 × 10^21^ (n/cm^2^)
Total (Full range)	4.05 × 10^21^ (n/cm^2^)
Gamma	7.23 × 10^21^ (gamma/cm^2^)

**Table 5 sensors-21-06094-t005:** Performance metrics for the bismuth titanate (BiTi) sensor. The fast fluence is calculated for the last measured acoustic response of the sensor considering neutrons greater than 1 MeV.

Fast Fluence (n/cm^2^)	% Change in Amp	% Change in f_c_
5 × 10^20^	–80%	0.0%

## Data Availability

Reinhardt, B. Nonlinear Ultrasonic Measurements in Nuclear Reactor Environments. Ph.D. Thesis, Pennsylvania State University, Stecker Ridge, PA, USA, 2016.

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
