# Peer review of "Use of the Ferroelectric Ceramic Bismuth Titanate as an Ultrasonic Transducer for High Temperatures and Nuclear Radiation"

_sensors, 2021, doi:10.3390/s21186094_

Round 1
Reviewer 1 Report
The manuscript is intended as a short review of work concerning bismuth titanate (author please note: chemical element names conventionally do NOT start with a capital letter).
Over 50% of the paper reports on experiments to assess the temperature and radiation hardness of bismuth titanate as part of an ultrasound transducer. The results are taken from the PhD thesis of Reinhardt (ref 9) conducted at Penn State University, presumably under the supervision of the author. Both the presentation of data and the conclusions are somewhat confusing.The data for the temperature dependence tests show poor reproducibility and there is no attempt to determine whether this is due to the intrinsic behavior of the material or poor experimental design, although the latter seems to be inferred. There is a clear presentational error associated with Figure 6 as it is referred to in two different contexts on page 5. In the first paragraph it is correctly referenced as amplitude of the power density vs neutron flux, however, it should be noted that a factor of 1020 is missing from the x-axis label. In the second paragraph, Fig 6 is discussed as if it were a plot of pulse echo amplitude vs time. Hence, there is a figure missing, which needs to be added during a revision or this second paragraph should be omitted. However, the two contexts appear to be supplementary rather than complementary.
The reference to the drop-in sample of bismuth titanate is also confusing. Was the sample electroded prior to neutron exposure and how was the testing carried out. The statement that "the impedance was measured at 11 kOhm", is out of context without a pre-exposure value to which to compare it and seems irrelevant at this point in the manuscript. The remaining 40% of the manuscript is taken up with very brief reports of other research on bismuth titanate either at high temperature or under radiation. This section lacks structure and does not seem to be arranged to provide the author with a means to provide a critical review of the content. as such, the adds little to the existing literature as it lacks the critical or comparative approach. i suspect it is lifted from the literature review of Reinhardt's thesis without much modification.
In summary, I feel that his paper could only be published in a highly modified form. It is currently neither one thing nor the other; if the intent was to provide an outlet for the work of Reinhardt, the first section fails as a competent publication in its own right for the reasons expressed above. As a review, it is neither sufficiently comprehensive nor rigorously critical to be of use to the interested reader.
Reviewer 2 Report
This paper aims to provide a perspective on the state-of-the-art of knowledge on Bismuth Titanate (BiTi) and its prospects for use as sensor for reactors at high temperature. The Modified or Augmented BiTi concerning to the temperature capability was also discussed in detail. These results offer a perspective on the state-of the-art for a possible sensor for harsh environments of high temperature and/or Gamma radiation and nuclear fluence. The following issues should be in-cooperated for the improvement of the present manuscript.
- In abstract, the chemical formula for lead-zirconate-titanate and bismuth titanate should be given as lead-zirconate-titanate (Pb(Zr, Ti)O3, PZT) and Bismuth Titanate (Bi4Ti3O12, BiTi).
- In the full text, the author uses “Bismuth Titanate (BiTi)”, “Bismuth Titanate Bi4Ti3O12 (BiTi)”, “Bismuth Titanate (Bi4Ti3O12) ”, “BiTi”, “Bi4Ti3O12”, and “Bismuth Titanate”. It makes the text look messy.
- The discussion on the material properties of Bismuth Titanate is necessary and should be enriched.
- Some abbreviations are not given full names in the text, such as Pz46, BT-100, KBT. This may damage the readability of the article and should be carefully addressed. More references should be provided for the benefit of readers to better understand the key concepts from material compositions, necessary processing to devices design, properties and applications.
Round 2
Reviewer 1 Report
I appreciate the scale of changes made by the author in response to the first review, however I still have a number of reservations about this manuscript. Much of the new text has been copied directly from the thesis of Reinhardt - has he granted permission to use his copyrighted material ? If not, the article might be considered to plagiarise the thesis. Why is Reinhardt not an author ?
In copying the text there are many formatting errors, particularly with respect to subscripts and superscripts.
There also appears to be redundant material, e.g. Figure 2.2. (which is actually a Table) provides lots of detail on two materials which are not part of the research, yet the material on which the research is based (BiT - TRS BT100 ?) is not presented with the same detail.
Is it necessary to include a photo of a piece of off-the-shelf electronics (Fig 7.1) ?
Author Response
Response to Reviewer #1
I appreciate the scale of changes made by the author in response to the first review, however I still have a number of reservations about this manuscript. Much of the new text has been copied directly from the thesis of Reinhardt - has he granted permission to use his copyrighted material ? If not, the article might be considered to plagiarise the thesis. Why is Reinhardt not an author ?
Response: Although Brian Reinhardt had previously refused to be a co-author because of time and work constraints, I have been able to convince him to become a co-author, as is certainly appropriate.
In copying the text there are many formatting errors, particularly with respect to subscripts and superscripts.
Response: The text has been modified extensively with respect to subscripts and superscripts.
There also appears to be redundant material, e.g. Figure 2.2. (which is actually a Table) provides lots of detail on two materials which are not part of the research, yet the material on which the research is based (BiT - TRS BT100 ?) is not presented with the same detail.
Response: Figure 2.2 has been eliminated and replaced by a Table which lists the most relevant material properties of BiTi-100 as displayed in Dr. Reinhardt’s thesis and in the Table of Properties listed in the brochure of by Del Piezo Specialties, LLC which was the original source of the material.
Is it necessary to include a photo of a piece of off-the-shelf electronics (Fig 7.1) ?
Response: The photo of the off-the-shelf electronics has been eliminated.
